# The Construction of ceRNA Regulatory Network Unraveled Prognostic Biomarkers and Repositioned Drug Candidates for the Management of Pancreatic Ductal Adenocarcinoma

**DOI:** 10.3390/cimb47070496

**Published:** 2025-06-27

**Authors:** Busra Aydin, Keziban Okutan, Ozge Onluturk Aydogan, Raghu Sinha, Beste Turanli

**Affiliations:** 1Department of Bioengineering, Faculty of Engineering and Architecture, Konya Food and Agriculture University, Konya 42700, Turkey; 2Department of Biotechnology, Institute of Postgraduate Education, Konya Food and Agriculture University, Konya 42700, Turkey; keziban.okutan.1@gmail.com; 3Department of Bioengineering, Faculty of Engineering, Marmara University, Istanbul 34854, Turkey; ozge.onluturk@gmail.com (O.O.A.); beste.turanli@marmara.edu.tr (B.T.); 4Department of Molecular and Precision Medicine, Penn State College of Medicine, Hershey, PA 17033, USA

**Keywords:** transcriptomics, circular RNAs, microRNAs, competing endogenous networks, drug repositioning, network signatures, prognostic biomarkers

## Abstract

Pancreatic ductal adenocarcinoma (PDAC) is one of the most lethal cancer types due to its late diagnosis, low survival rates, and high frequency of metastasis. Considering the molecular mechanism of PDAC development has not been fully elucidated, this study aimed to shed more light on the molecular regulatory signatures of circular RNAs (circRNAs) in PDAC progression and provide a different perspective to identify potential biomarkers as well as discover candidate repositioned drug molecules for the prevention or treatment of PDAC with network-based integrative analysis. The mRNA, miRNA, and circRNA expression profiles of PDAC were obtained from nine microarray datasets. Differentially expressed genes (DEGs), microRNAs (DEmiRNAs), and circular RNAs (DEcircRNAs) were identified. The competing endogenous RNA (ceRNA; DEG–DEmiRNA–DEcircRNA) regulatory network was constructed, which included 12 DEcircRNAs, 64 DEGs, and 6 miRNAs specific to PDAC. The *ADAM12*, *MET*, *QKI*, *SEC23A*, and *ZEB2* were identified as hub genes and demonstrated significant survival probability for PDAC. In addition to providing novel biomarkers for diagnosis that can be detected non-invasively, the secretion levels of hub genes-associated proteins were found in plasma, serum, and oral epithelium. The drug repositioning analysis revealed vorinostat, meclocycline sulfosalicylate, and trichostatin A, which exhibited significant binding affinities to the hub genes compared to their inhibitors via molecular docking analysis.

## 1. Introduction

Pancreatic cancer is one of the diseases with the worst prognosis [1]. It ranks as the sixth leading cause of cancer deaths in both sexes combined [2]. Pancreatic cancer is responsible for 5% of all cancer mortality worldwide [2]. In 2022, 511,000 new cases of pancreatic cancer were diagnosed, and 467,000 deaths occurred [2]. Pancreatic ductal adenocarcinomas (PDAC) are among the most prevalent exocrine cell tumors, accounting for approximately 95% of cases of pancreatic cancer [3]. It is important to differentiate between advanced chronic pancreatitis, primary pancreatic neoplasms, and ductal adenocarcinoma [4]. In contrast to histological criteria of chronic pancreatitis, which is characterized by regular, lobular formations, intrapancreatic localization, intact ducts, and preserved nuclear polarity, PDAC exhibits characteristics such as irregular distribution, perineural and extrapancreatic dissemination, nuclear pleomorphism, and commonly lost nuclear polarity [4]. The various subtypes of PDAC are classified as ductal adenocarcinoma, adenosquamous carcinoma, colloid carcinoma, anaplastic carcinoma, sarcomatoid carcinoma, and osteoclastic giant cell carcinoma, according to the World Health Organization [5]. The reasons for the poor prognosis of PDAC include delayed diagnosis, lack of effective drugs, and/or resistance of neoplastic cells to conventional chemotherapy [6,7].

Circular RNA (circRNA) is one of the most significant and highly stable single-stranded noncoding RNAs (ncRNAs) because of its covalently closed loop structure [8]. circRNAs are endogenous biomolecules that lack both 5′ end caps or 3′ poly(A) tails [9]. The stability of circRNAs makes them crucial for the development and diagnosis of human diseases and malignancies [10]. circRNAs serve as molecular sponges that sequester microRNAs (miRNAs) from interacting with their target messenger RNAs (mRNAs) [11]. They can control the availability and activity of miRNAs through the sponging mechanism, which impacts many cellular pathways and activities, implicating the development of cancer [12]. For example, circRNA transcriptional adaptor 2A (circTADA2A) has been shown to modulate the microRNA-203a-3p/suppressor of cytokine signaling 3 (SOCS3) axis in breast cancer, thereby suppressing cell proliferation, migration, and invasion [13]. In another example, circ-0001823 suppresses miR-613, which influences RAB8A expression and encourages proliferation and metastasis in cervical cancer [14]. circRNAs control the epigenetic gene expressions through miRNAs, which establish a circRNA–miRNA–mRNA network, known as competing endogenous RNA (ceRNA) [15].

Despite effortful studies having been conducted to understand the mechanism behind the pathogenesis of PDAC, as well as the suggestion of several small molecules for treatment of the disease, there is an unmet need for the development of omics-oriented drugs. Although several signs of progress have been made in the treatment of pancreatic cancer [16,17,18], clinical trial data are still largely missing and are urgently needed to verify relevant effects and for the development of more personalized treatment approaches. This study compared the microarray datasets of PDAC patients as well as controls for the differentially expressed circRNAs, miRNAs, and mRNAs (DEcircRNAs, DEmiRNAs, and DEGs). Furthermore, a detailed investigation of the “DEcircRNA–DEmiRNA–DEGs”-competing endogenous RNA (ceRNA) network was performed for PDAC to reveal novel disease signatures (Figure 1). Several potential repositioned drugs for PDAC were identified via hub gene–drug interaction, and molecular docking analysis was used to validate these in silico. Altogether, this investigation improves the current understanding of the ceRNA regulatory network and the mechanisms involved in the regulation of PDAC carcinogenesis and provides novel candidate biomarkers for prognosis and therapeutic agents.

## 2. Materials and Methods

### 2.1. Selection of Microarray Datasets

Data from PDAC and healthy tissues were obtained from the National Center for Biotechnology Information (NCBI) Gene Expression Omnibus (NCBI-GEO) database [19]. These datasets included four mRNA expression profiles (GSE15471, GSE41368, GSE71989, and GSE62165), three miRNA expression profiles (GSE60978, GSE32678, and GSE41369), and two circRNA expression profiles (GSE69362 and GSE79634) (Table 1).

### 2.2. Identification of DEGs, DEmiRNAs, and DEcircRNAs

The mRNA, miRNA, and circRNA datasets were analyzed using the online statistical tool GEO2R (https://www.ncbi.nlm.nih.gov/geo/geo2r/, accessed on 10 February 2025) [19], which combines R/Bioconductor 4.2.2, Biobase 2.58.0, GEOquery 2.66.0, and Linear Models for Microarray Analysis (limma) 3.54.0. The expression values were normalized using the quantile normalization method, which is embedded in the limma package. The Benjamini–Hochberg method was used to control the false discovery rate. To acquire the differentially expressed genes (DEGs), the differentially expressed miRNAs (DEmiRNAs), and the differentially expressed circRNAs (DEcircRNAs) from the dataset, the cut-off criterion for |log_2_(fold change)| > 1 and *p*-value < 0.05 were determined for all datasets. The directions of expressions were designated as follows: log_2_FC ≤ −1 as downregulated and log_2_FC ≥ 1 as upregulated. To identify common DEGs, DEmiRNAs, and DEcircRNAs, Venn diagrams were constructed for expression profiles separately through Jvenn (https://jvenn.toulouse.inrae.fr/app/example.html, accessed on 17 February 2025) [31].

### 2.3. Functional Enrichment Analysis

The functional enrichment analysis for common DEGs was performed using GeneCodis v4 [32]. The Kyoto Encyclopedia of Genes and Genomes (KEGG) [33] database and the Gene Ontology (GO) [34] terminology were used for pathway analysis and determination of biological processes, cellular processes, and molecular functions. Results with *p* < 0.05 were considered statistically significant.

### 2.4. Construction of ceRNA Network

The relationships between common DEcircRNAs and DEmiRNAs were found in data downloaded from the circBank [35], an extensive human circRNA database. Afterwards, data was downloaded from miRTarBase [36] to construct interactions between common DEmiRNAs and DEGs. The ceRNA (DEGs–DEmiRNAs–DEcircRNAs) regulatory network was constructed by combining DEcircRNA–DEmiRNA pairings with DEmiRNA–DEG pairs. Then, Cytoscape (3.9.1) [37] was used to depict the ceRNA regulatory network. The topological analysis of the network was conducted using both local and global metrics including degree and betweenness centrality features. Using Metascape (v3.5.20240901), GO and KEGG pathway analyses were performed on target DEGs in the ceRNA regulatory network.

### 2.5. Survival Analysis

Cox regression was performed to elucidate survival-associated genes. The results were visualized and compared through Kaplan–Meier (KM) plots using the follow-up times of high-risk or low-risk groups. The prognostic power of the module was determined using the log-rank test *p*-value, the hazard ratio (HR), and its confidence intervals (CIs). Survival analysis was performed with the DEGs in ceRNA network construction through the KM-Plotter tool (https://kmplot.com/analysis/index.php?p=service, accessed on 5 March 2025) [38]. The log-rank *p*-value < 0.05 was accepted as statistically significant.

### 2.6. Determination of Secretion Levels of Hub Genes-Associated Proteins in Different Tissues

The secretion levels (ppm) of common hub genes-associated proteins were obtained through protein expression data available in the GeneCards database [39]. GeneCards is periodically curated and integrates the protein expression data from external proteomics databases such as ProteomicsDB [40], MaxQB [41], and MOPED [42]. The different tissues/biological fluids such as the pancreas, pancreatic juice, plasma, serum, urine, oral epithelium, and saliva were used, and these were investigated through protein expression data accessible in the above databases.

### 2.7. Drug Repositioning

Potential drugs and small compounds based on the expression signatures of target DEGs were repositioned through the Library of Integrated Network-based Cellular Signatures (LINCS)–L1000CDS^2^ data [43]. The resultant repositioned drugs were selected based on their cosine distance scores (1-cos α), the Food and Drug Administration (FDA) approval status, and limitations. The cosine distance search allowed us to perform a similarity search among input genes when these were exposed to candidate repositioned drugs. The largest cosine distance scores represented reverse interactions between input genes and candidate drugs. The mechanism of action and indications of these drugs were cross-checked through published literature and publicly available datasets such as PubChem [44], Comparative Toxicogenomics Database (CTD) [45], and Drug Bank [46].

### 2.8. Molecular Docking

The three-dimensional (3D) structures of the target DEGs of DEcircRNAs were obtained from the Protein Data Bank (PDB) [47] and UniProt [48]. Inhibitors of each prognostic biomarker were identified from the CTD [45], and the 3D structures of potential drugs and inhibitors were acquired from PubChem [44]. Molecular docking analyses were performed using the Cavity detection-guided Blind Docking (CB-Dock v2) [49] online tool.

## 3. Results

### 3.1. Identification of DEGs, DEcircRNAs, DEmiRNAs, and Enrichments of DEGs

A total of 817 DEGs were identified from four sets of mRNA expression profiles (GSE15471, GSE41368, GSE71989, and GSE62165). A total of 37 DEcircRNAs were found in circRNA expression profiles (GSE69362 and GSE79634), and 7 DEmiRNAs were obtained in miRNA expression profiles (GSE60978, GSE32678, and GSE41369) according to the thresholds as indicated in Venn diagrams (Figure 2).

The enrichment analysis revealed distinctive biological characteristics and molecular functions of the common DEGs. According to pathway analysis, the DEGs were enriched in the phagosome, complement, and coagulation cascades, *Staphylococcus aureus* infection, focal adhesion, leishmaniasis, extracellular matrix (ECM)–receptor interaction, amoebiasis, cytoskeleton in muscle cells, toxoplasmosis, and viral myocarditis. The GO analysis highlighted significant enrichment in cell adhesion, ECM, virus response, angiogenesis, cell migration regulation, endodermal cell differentiation, defense response to virus, collagen fibril organization, and immune response (Figure 3A,B). The functional annotations of the DEmiRNAs were significantly enriched with positive regulation of osteoblast differentiation (Figure 3C). DEmiRNAs were enriched in cancer-related pathways such as microRNAs and proteoglycans in cancer and the chemical carcinogenesis receptor activation pathway (Figure 3D).

### 3.2. Visualization of ceRNA Network

The PDAC-specific differential expression network was constructed using the interactions of common 64 DEGs, 6 DEmiRNAs, and 12 DEcircRNAs. This network revealed 80 nodes and 138 links between differentially expressed elements (Figure 4A). The topological analysis of the ceRNA network was carried out based on the local and global metrics including degree and betweenness centrality. The significant hub elements (MET, SEC23A, STC2, ZEB1, ZEB2, hsa-miR-130b-5p, hsa-miR-135b-3p, hsa-miR-135b-5p, hsa-miR-199a-3p, hsa-miR-200c-3p, hsa-miR-200c-5p) are depicted in Figure 4B. This network represents the topologically significant hub elements and differs from the previous network by interactions with 79 links around 80 nodes. It was named the ceRNA expression network of the PDAC. To narrow down the interaction list and elucidate the most specific subnetwork composed of nodes that are more closely related than the rest of the components, we constructed a PDAC-specific core network (Figure 4C). This network represents the most likely small circuits preferred by cells in PDAC pathogenesis regarding information flow. The PDAC-specific core network was composed of 15 nodes and 14 edges, and ADAM12, MET, QKI, THBS2, SEC23A, ZEB1, and ZEB2 were identified as hub elements of this network (Figure 4C). Also, there were circRNAs, including hsa_circRNA_102465 and hsa_circRNA_100904, that act as signal distributors of the PDAC-specific core network. The miRNAs such as hsa-miR-135b-5p, hsa-miR-200c-5p, hsa-miR-199a-3p, hsa-miR-130b-5p, hsa-miR-135b-3p, and hsa-miR-200c-3p were also identified in the PDAC-specific core network.

### 3.3. Survival Analysis and Subnetwork Construction of ceRNA

The KM-Plotter tool uses gene expression data, relapse-free, as well as overall survival information from the most commonly preferred sources of high-throughput datasets such as Gene Expression Omnibus (GEO), European Genome–Phenome Archive (EGA), and The Cancer Genome Atlas (TCGA). To analyze the prognostic power of an individual gene, the patient samples are divided into two groups according to varied quantile expressions of the proposed biomarker. The survival probabilities were compared using the KM plots of the hub gene expressions (Figure 5) between the two different groups to determine prognostic biomarkers. Among these, 5 hub genes ADAM12 (HR = 1.28, *p* = 9.29 × 10^−4^), MET (HR = 1.77, *p* = 4.78 × 10^−4^), QKI (HR = 1.16, *p* = 1.33 × 10^−4^), SEC23A (HR = 1.23, *p* = 0.00548), and ZEB2 (HR = 1.24, *p* = 0.00561) were identified and these had higher expression and significant correlation with overall survival time. The prognostic biomarkers that demonstrated an effect on PDAC patient survival are listed in Table 2.

### 3.4. Secretion Levels of Hub Genes-Associated Proteins in Different Tissues

QKI protein is expressed highly in both plasma and pancreas, and their expression levels are comparable. MET is expressed in the pancreas; however, the expression level is significantly higher in serum. In contrast, SEC23A protein has elevated levels in the pancreas and it is also expressed in oral epithelium.

### 3.5. Drug Repositioning with ceRNA Subnetwork

Drug repositioning was carried out through L1000CDS^2^ depending on 5 hub genes and their fold change values. While circRNAs were crucial in the construction of the ceRNA network and the elucidation of upstream regulatory pathways, our subsequent analyses focused on mRNAs for drug repositioning purposes. This decision was methodologically driven, as current drug repositioning platforms—such as L1000CDS^2^ and CMAP—operate primarily at the mRNA level and are not yet optimized for circRNA-based screening. Therefore, we prioritized mRNAs that are central nodes within the ceRNA network and involved in relevant signaling pathways as therapeutic targets. Nevertheless, circRNAs played a foundational role in identifying these mRNAs within the regulatory network, and their expression profiles were considered in the mechanistic inference.

After using 5 hub genes as a query, the 50 drugs that could reverse the PDAC scenario were suggested via the search engine, and they were eliminated depending on their approval status, mechanism of action, indications, and antineoplastic activities. Among these 50 drugs, 4 drugs (vorinostat, trichostatin A, meclocycline sulfosalicylate, and guanabenz acetate) were selected, and 1 of them was approved, whereas 3 drugs were in the investigational status according to the FDA. Table 3 summarizes the detailed properties of the four repurposed drugs.

### 3.6. Molecular Docking Analysis with Candidate Repositioned Drugs

The interactions of selected drugs (vorinostat, trichostatin A, meclocycline sulfosalicylate, and guanabenz acetate) with target hub genes of PDAC were analyzed through the molecular docking method (Figure 6). The inhibitors of the target hub genes are abrine (ADAM12), acetaminophen (MET), aristolochic acid I (QKI), 1,2-Dimethylhydrazine (SEC23A), and antimycin A (ZEB2), which were considered as the control group, and their binding affinities (kcal/mol) were compared against the selected drugs. Accordingly, trichostatin A and meclocycline sulfosalicylate had the highest binding affinities with 80% (*n* = 4 out of 5 hubs) compared to vorinostat (60%, *n* = 3) and guanabenz acetate (40%, *n* = 2). The binding affinities of each target hub gene with repurposed drugs were ADAM12 at 75%, MET at 100%, QKI at 0%, SEC23A at 100%, and ZEB2 at 50% compared to the control group. MET and SEC23A were the most significant hubs among the target hub genes regarding molecular docking results.

### 3.7. Cross-Validation of Protein and Gene Expressions of Proposed Diseased Signatures

We systematically compared expression profiles between diseased and healthy tissues, leveraging publicly available high-resolution transcriptomic and proteomic datasets. Specifically, we assessed the spatial and cell-type-specific expression of the proposed marker genes across multiple tissue types (including normal and PDAC tissues), ensuring a comprehensive validation framework. This analysis provides crucial mechanistic insights by establishing the differential expression patterns that underpin disease-associated molecular signatures. In the immunohistochemical (IHC) stainings, the proposed biomarkers demonstrated distinct protein expression values in PDAC tissues (Figure 7A). The proposed disease signatures exhibit significantly altered gene expression in diseased tissues compared to their healthy counterparts, consistent with their hypothesized role in disease pathogenesis (Figure 7B). The study employed a systems biology-driven approach that integrates transcriptomic data and network-based inference to prioritize candidate targets. Such computational predictions inherently reflect probabilistic associations rather than deterministic outcomes. Discrepancies between mRNA and protein expression are not uncommon, as they can result from various layers of post-transcriptional, translational, or post-translational regulation, as well as tissue-specific expression dynamics and technical limitations of detection methods such as IHC.

## 4. Discussion

PDAC is the most prevalent pancreas malignant neoplasm [69] and the most lethal among solid malignancies [70]. PDAC is a heterogeneous neoplasm that exhibits varied morphologic and genetic features as well as a poor clinical prognosis [69]. Cytotoxic chemotherapy continues to be the cornerstone of systemic therapy for PDAC, making treatment a significant challenge [71]. Nevertheless, the available first-line therapies might have serious side effects, and there is an urgent need for new treatment solutions. Today, the field of ‘omics’ uses systems-oriented approaches to better understand the predictive and prognostic phenotypes of various diseases, including cancer [72,73,74].

circRNAs have been observed in the initiation and progression stages of various malignancies [75,76]. These circRNAs also have the ability to target miRNAs, which are known as “miRNA sponges”, in order to decrease their levels and remove their target mRNA inhibitions, which in turn controls the gene expressions that code for proteins [77,78]. The primary function of circRNA in different types of cancer is thought to be the sponging of miRNA by ceRNA [79]. The majority of research has been focused on how circRNAs control the growth of tumors through miRNAs [80,81,82].

In this study, we constructed a ceRNA network related to circRNA by thoroughly analyzing the mRNA, miRNA, and circRNA expression patterns of PDAC profiles. The ceRNA network comprised 12 DEcircRNAs (Appendix A), 64 DEGs (Appendix A), and 6 DEmiRNAs. Depending on the ceRNA network, circRNAs and mRNAs function as ceRNAs and have the same miRNA binding sites. Our results indicated that hsa_circRNA_102465 and hsa_circRNA_100904 have the most miRNA interactions, which might be crucial for the progression of PDAC.

Regarding our findings, several hub genes, including ADAM12, MET, THBS2, ZEB1, and ZEB2, have been previously implicated in PDAC progression, tumor invasiveness, and epithelial–mesenchymal transition (EMT). ZEB1 and ZEB2 are established transcriptional repressors of E-cadherin and are well-documented drivers of EMT and metastasis in PDAC [83,84]. Yang et al. showed that miR-135-5p expression increased in PDAC patients [85]. A study conducted in 2023 reported an interaction between miR-135-5p and the ADAM12 gene [86]. miR-135-5p expression was detected in placenta and was associated with preeclampsia, and could be used as a potential molecular marker. Furthermore, the elevated miR-135-5p levels interact with the ADAM12 gene and prevent migration and invasion in trophic cells [86]. In another report, miR-135-5p was shown to inhibit cell proliferation by targeting the ADAM12 gene in glioma cells [87,88].

Another candidate, miRNA-200c, has been shown to associate with the metastasis of various tumors [89] and prevent the invasion of tumor tissue in melanoma, breast, and pancreatic cancers [90,91,92]. It has been reported that miRNA-200c closely interacts with the ZEB1 and ZEB2 proteins and prevents invasion in cancer cells through this complex [93,94]. It has also been shown to be effective in directing the EMT mechanism, which is crucial for the metastasis of tumor cells and the progression of cancer [95].

MET proto-oncogene is negatively regulated by miRNA-199a. It is thought that there is potential for preventing the metastasis and progression of cancer cells inhibiting the MET signaling pathway and inducing apotosis in several cancer types [96].

In parallel, several of the identified miRNAs, such as miR-200c-5p, miR-135b-5p, and miR-130b-5p, have been shown to regulate these hub genes in PDAC or related cancer types. miR-200c-5p, for instance, is a well-characterized suppressor of ZEB1/ZEB2, and its downregulation is associated with EMT activation in pancreatic cancer [97]. Similarly, miR-135b-5p has been linked to the regulation of MET and ADAM12, affecting invasion and proliferation in multiple cancers including PDAC [98,99].

Although functional studies on circRNAs such as hsa_circRNA_102465 and hsa_circRNA_100904 remain limited, computational predictions and prior expression data suggest their potential involvement in gastrointestinal cancers and their capability to act as sponges for miRNAs relevant to PDAC regulatory circuits [100].

The protein expression levels in different tissues play a key role in the early diagnosis, monitoring, and treatment of diseases. Elevated or reduced levels of these proteins could be used as a biomarker of some diseases like cancer. They provide information about the presence, progression, or response to treatment. For this reason, we performed an analysis to discover the secretion level of hub genes-associated proteins in different tissues. In this context, the protein expressions in plasma, serum, urine, and saliva could provide an opportunity for diagnosis and response to treatment of PDAC. The significantly high levels of MET, QKI, and SEC23A expressions could be utilized as novel biomarkers for the early diagnosis, monitoring, and effective treatment of PDAC. Also, supportive information in the literature shows that signaling from MET, the receptor for hepatocyte growth factor, promotes pancreatic tumorigenesis and poor patient prognosis by enhancing tumor cell growth, survival, and motility [101]. QKI has been identified as a splicing signature that can distinguish basal-like PDAC subtypes and may predict worse clinical outcomes in pancreatic cancer [102]. SEC23A has been indicated as a potential prognostic marker in bladder cancer [103]. In addition, other probable markers like ADAM12 have been reported as a circulating marker for stromal activation in pancreatic cancer and could predict responses to chemotherapy [104]. Moreover, THBS2 was recently evaluated as a prognostic marker in PDAC metastatic patients [105]. However, it is critical to find these proteins in tissues/biofluids where biological samples can easily be obtained, thus providing a non-invasive method of diagnosis.

Drug repositioning was carried out regarding five hub genes *ADAM12*, *MET*, *QKI*, *SEC23A*, and *ZEB2*. This approach provides a significant advantage for identifying novel pancreatic drugs by decreasing drug development costs and time. Among the 50 drugs, 4 drugs (vorinostat, trichostatin A, meclocycline sulfosalicylate, and guanabenz acetate) were selected to reverse PDAC expression depending on the hub genes’ drug repositioning (Table 3). The drug repositioning approach employed in our study was based on transcriptomic reversal signatures using the L1000CDS^2^ platform. This method does not rely on the traditional mechanism of action alignment but instead uses the ability of small molecules to reverse disease-specific gene expression patterns. Accordingly, the identification of an antimicrobial drug, an antihypertensive agent, and histone deacetylase inhibitors as repositioned drug candidates was not based on their function per se, but rather on their predicted capacity to modulate the expression of oncogenic hub genes implicated in our network analysis. The docking analyses were performed with these drugs to verify results and the binding affinities to target hub genes (*ADAM12*, *MET*, *QKI*, *SEC23A*, and *ZEB2*) via in silico analyses.

Vorinostat (Zolinza^®^, Merck Research Laboratories) belongs to an inhibitor of the histone deacetylase (HDAC) drug class [64]. It is an FDA-approved drug and is used for cutaneous T-cell lymphoma treatment [64]. The HDAC enzyme family catalyzes the removal of acetyl groups from the lysine residues of histone tails, thus resulting in a more compacted chromatin [106]. HDAC inhibitors have a variety of particular impacts on distinct cell types both in vivo and in vitro, including stopping the proliferation of malignant cells, altering their differentiation, regulating migration and angiogenesis, and triggering their apoptosis [107]. Furthermore, BCL6, E2F1, GATA1, P53, SMAD7, and YY1 are transcriptional factors that regulate the genes involved in cell cycle progression, apoptosis, and immune response, and can be directly or indirectly affected by vorinostat [108,109,110,111,112]. Vorinostat was also effective against parasitic diseases such as amoebiasis, cryptosporidiosis, leishmaniasis, malaria, and schistosomiasis [113]. It has been indicated that vorinostat has antiproliferative activity in breast cancer [114], ovarian or primary peritoneal carcinoma [115], lung cancer [116], and gastric cancer [117].

Trichostatin A (TSA) inhibits a hydroxamic acid Class I HDAC inhibitor (HDAC1, 2, 3), Class IIa HDAC inhibitor (HDAC4, 7, 9), and Class IIb inhibitor (HDAC6). It has antineoplastic, antibacterial, and antifungal properties [66]. TSA was first identified as an antifungal antibiotic, isolated from a *Streptomyces platensis* culture broth [118]. TSA can induce differentiation, cell cycle arrest (G1 and G2 phases), and reverse transform cell morphology to a more normal state in culture [119]. Platta et al. indicated that TSA inhibits the proliferation of small-cell lung cancer cells by inducing morphological differentiation, cell cycle arrest, and apoptosis, which inhibits cell growth in a dose-dependent manner [120]. TSA has also been investigated in PDAC in vitro [121].

Meclocycline sulfosalicylate is a tetracycline, a class of antibiotics used to treat various bacterial infections such as acne vulgaris [122,123]. It binds to the 30S ribosomal subunit and inhibits protein synthesis in bacteria [67]. Thus, adding amino acids to the growing peptide chain is hindered. Also, a wide range of Gram-positive and -negative bacteria are susceptible to the action of meclocycline sulfosalicylate [67].

Guanabenz acetate (GA, Wytensin^TM^) was identified as an antihypertensive drug for the treatment of high blood pressure and functions as an α-2 adrenergic agonist [68]. It also shows anti-prion activity [124]. The mechanism of action of GA works by inhibiting the dephosphorylation of eukaryotic initiation factor-2 alpha (eIF2α) caused by stress and thus causing suppression of endoplasmic reticulum stress [125]. The Rac1-GTPase signaling pathway is downregulated due to increased eIF2α phosphorylation [126]. Overexpression of Rac1-GTPase was observed in triple-negative breast cancer [127].

According to docking results, trichostatin A, meclocycline sulfosalicylate, and vorinostat had significant binding affinities with the hub genes regarding their inhibitors. Moreover, *MET* and *SEC23A* were effectively docked with all five repurposed drugs.

*MET* is a proto-oncogene and the transmembrane tyrosine kinase receptor for scatter factor/hepatocyte growth factor (SF/HGF), which are pleiotropic cytokines [128,129]. When HGF binds to the *MET* receptor, it triggers cell motility, division, survival, and differentiation [130]. *MET* has been linked to various biological processes, including tissue repair, morphogenesis, and embryonic and neural development [54].

The *SEC23A* gene is a main component of COPII, which controls the vesicle budding from the ER [59,131]. Face dysmorphisms, skeletal abnormalities, sutural cataracts, late-closing fontanels, and other symptoms are the main characteristics of cranio-lenticulo-sutural dysplasia (CLSD), which is caused by two missense mutations in *SEC23A* [60,132]. Zhu et al. indicated the complete absence of *SEC23A*, which causes it to be fatal during mid-embryogenesis [133]. The molecular mechanism(s) for the role of suggested prognostic markers as well as the impact of drug combinations will be studied in the future, using in vitro experiments utilizing pancreatic ductal adenocarcinoma cells lines.

While the proposed drug candidates demonstrate promising therapeutic potential based on in silico analyses, their actual efficacy in biological systems remains contingent upon a range of pharmacokinetic and pharmacodynamic parameters—including absorption, distribution, metabolism, excretion, and toxicity (ADMET)—which necessitate rigorous evaluation through both in vitro and in vivo experimental models. Additionally, the biological relevance of these findings requires further validation. Specifically, corroborating the differential expression, the construction of the PDAC-specific ceRNA network and functional roles of these hub genes at both the transcriptomic (e.g., quantitative PCR) and proteomic (e.g., Western blotting, immunohistochemistry) levels in clinically relevant samples from PDAC patients would significantly enhance the translational value and scientific robustness of our conclusions.

Collectively, this study offers a novel integrative framework that combines circRNA-associated ceRNA network construction with transcriptomics-driven drug repositioning to uncover previously uncharacterized regulatory mechanisms and therapeutic opportunities in PDAC. Through the construction of a PDAC-specific ceRNA network, we identified key regulatory axes involving hsa_circRNA_102465, hsa_circRNA_100904, miR-135b-5p, miR-200c-5p, and miR-130b-5p, which interact with oncogenic drivers such as ADAM12, MET, QKI, SEC23A, and ZEB2. These interactions offer insight into the post-transcriptional regulation of key oncogenic drivers involved in EMT and tumor progression. Additionally, our use of the L1000CDS2-based transcriptomic reversal approach for drug repositioning represents an innovative strategy in PDAC, distinct from conventional mechanism-of-action-based repurposing. This enabled the identification of non-traditional candidates—some with no prior links to PDAC—such as meclocycline sulfosalicylate and guanabenz acetate, alongside the HDAC inhibitors vorinostat and trichostatin A. Taken together, these findings not only expand the current understanding of circRNA-mediated regulation in PDAC but also provide a promising foundation for non-invasive biomarker development and the preclinical evaluation of repositioned therapeutics. Altogether, these findings provide a strong rationale for further experimental validation and may contribute to the development of novel diagnostic and therapeutic strategies in PDAC.

## 5. Conclusions

This ceRNA network elucidated that hsa_circRNA_102465 and hsa_circRNA_100904 were crucial players in regulating the gene expression orchestra in PDAC as well as hub microRNAs, including hsa-miR-200c-5p, hsa-miR-199a-3p, hsa-miR-200c-3p, hsa-miR-130b-5p, hsa-miR-135b-3p, and hsa-miR-135b-5p. Under the effect of ceRNA regulation, cascades *ADAM12*, *MET*, *QKI*, *SEC23A*, and *ZEB2* were reported as PDAC disease signatures that significantly impacted the survival rates of pancreatic cancer patients. Based on these signatures, a drug repositioning analysis was conducted, which resulted in the identification of potential drug candidates that aimed to reverse the expression of these signatures to converge the healthy-state gene expressions. As a result, therapeutics such as trichostatin A, vorinostat, and meclocycline sulfosalicylate are being proposed as potential repositioned drug candidates for the therapeutic management of PDAC.

## Figures and Tables

**Figure 1 cimb-47-00496-f001:**
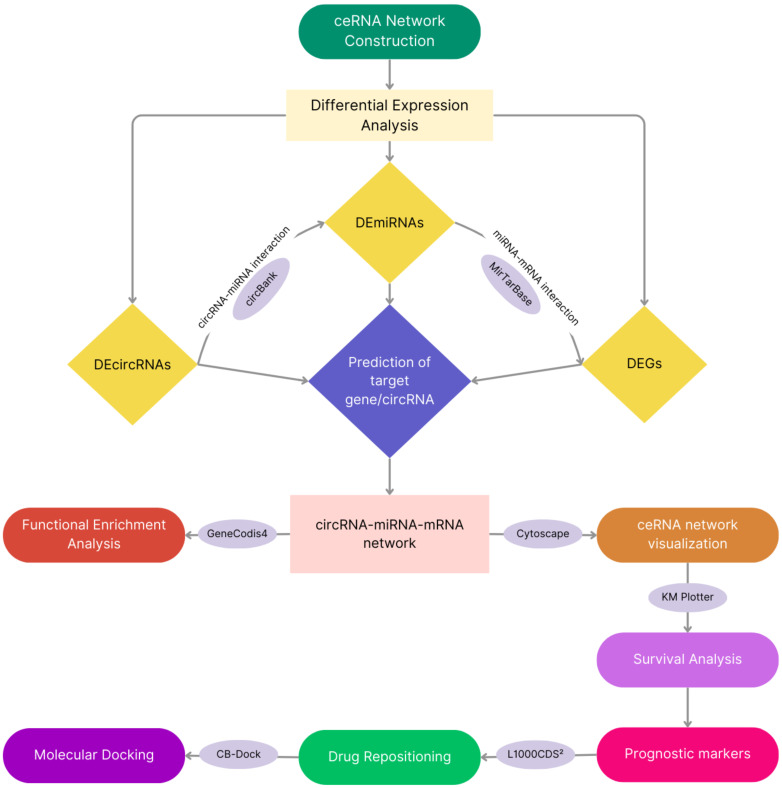
The framework for data collection and analysis in this study.

**Figure 2 cimb-47-00496-f002:**
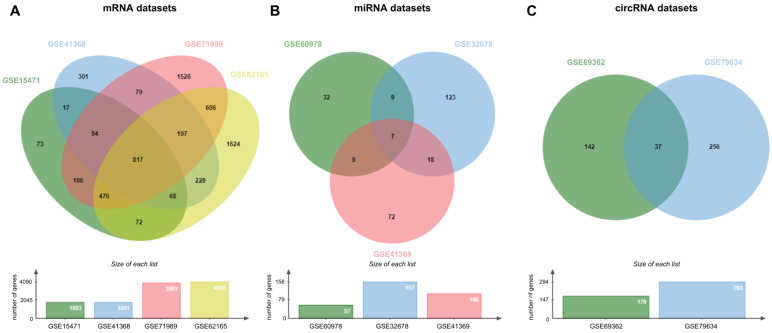
Identification of common (**A**) DEGs, (**B**) DEmiRNAs, and (**C**) DEcircRNAs between different datasets. To eliminate systemic bias, the expression values of all datasets were normalized with the quantile normalization method. The Benjamini–Hochberg method was used to control the false discovery rate. To identify the differentially expressed elements, the cut-off criterion was applied as |log2(fold change)| > 1 and *p*-value < 0.05 for all datasets.

**Figure 3 cimb-47-00496-f003:**
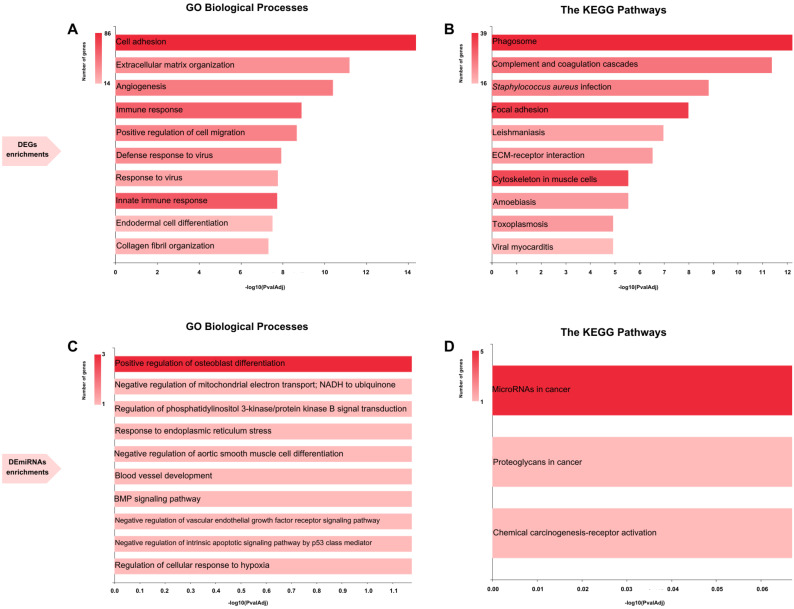
Functional enrichment analyses of DEGs in (**A**) GO biological processes and (**B**) KEGG pathways. DEmiRNAs’ functional enrichment analyses in (**C**) GO biological processes and (**D**) KEGG pathways.

**Figure 4 cimb-47-00496-f004:**
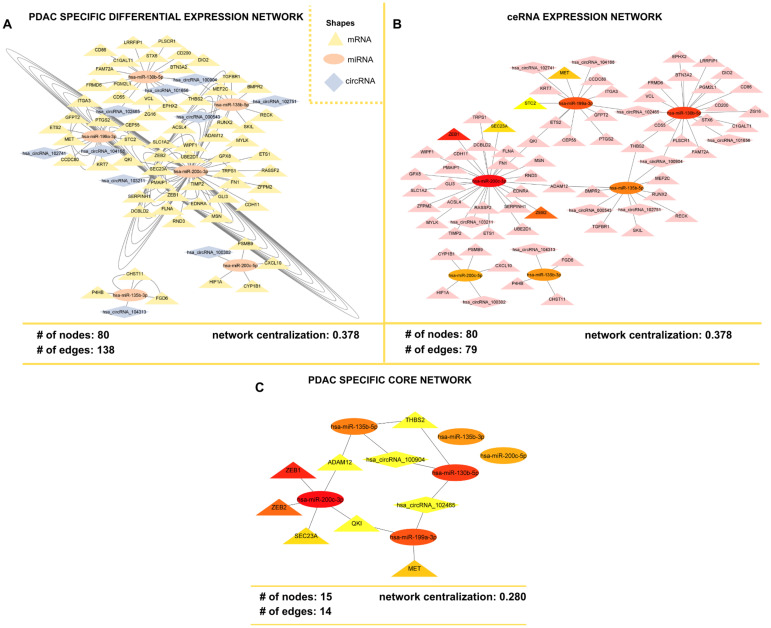
(**A**) PDAC-specific differential expression network, (**B**) ceRNA expression network, and (**C**) PDAC-specific core network.

**Figure 5 cimb-47-00496-f005:**
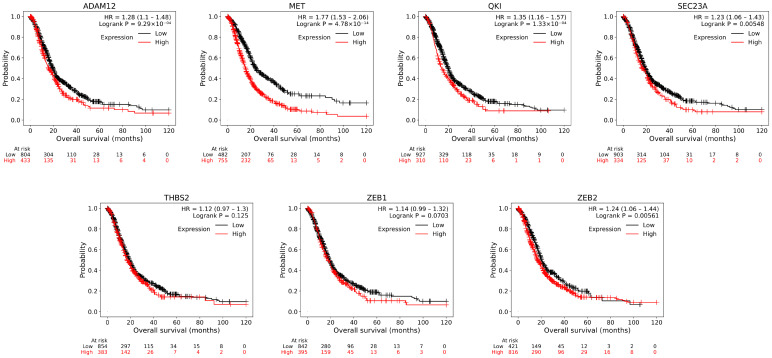
The overall survival curves for hub genes based on gene expression in the ceRNA network. Among the 7 hub genes, 5 of them (ADAM12, MET, QKI, SEC23A, and ZEB2) demonstrated significant survival probabilities. Survival curves were generated using the KM-Plotter tool, incorporating 1640 samples, including 1435 tumor tissues and 205 control tissues for 20,433 genes, as described by Posta and Győrffy [38]. The patient cohorts were provided from publicly available GEO and ICGC datasets, and 94% of the tumors were identified as PDAC. The log-rank test was used to assess overall survival (OS), and hazard ratios (HRs) with 95% confidence intervals were calculated. Log-rank *p*-values < 0.05 were considered statistically significant.

**Figure 6 cimb-47-00496-f006:**
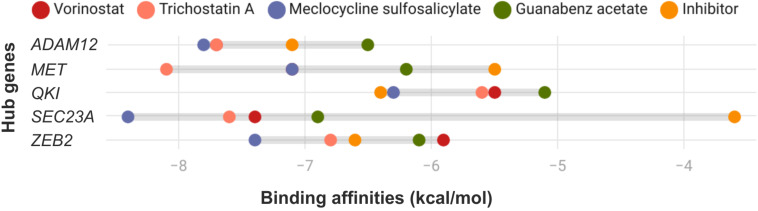
The binding affinities of hub genes with repurposed drug candidates and their specific inhibitors as revealed by molecular docking analysis.

**Figure 7 cimb-47-00496-f007:**
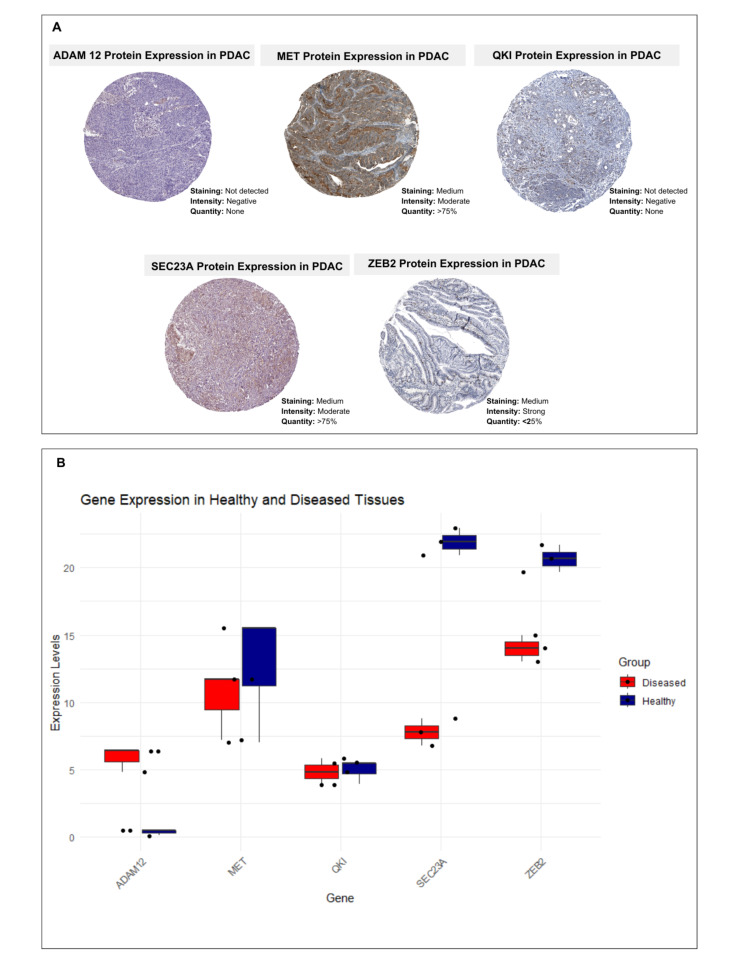
The protein (**A**) and gene (**B**) expression values of proposed PDAC biomarkers, which were collected from the Human Protein Atlas, were demonstrated.

**Table 1 cimb-47-00496-t001:** The dataset selection for the normal and PDAC pairs-specific transcriptome analysis.

Detected RNA Types	GEO Reference Series	Platform	Samples	Study Design	Ref.
Messenger RNA	GSE15471	Affymetrix Human Genome U133 Plus 2.0 Array	78	39 N/ 39 PDAC	[20,21]
	GSE41368	Affymetrix Human Gene 1.0 ST Array	12	6 N/ 6 PDAC	[22]
	GSE71989	Affymetrix Human Genome U133 Plus 2.0 Array	22	8 N/ 14 PDAC	[23]
	GSE62165	Affymetrix Human Genome U219 Array	131	13 N/ 118 PDAC	[24]
MicroRNA	GSE60978	Agilent-031181 Unrestricted Human miRNA V16.0 Microarray 030840	57	6 N/ 51 PDAC	[25]
	GSE32678	miRCURY LNA microRNA Array, v.11.0—hsa, mmu, and rno	32	7 N/ 25 PDAC	[26,27]
	GSE41369	NanoString nCounter Human miRNA assay V1	18	9 N/ 9 PDAC	[22]
Circular RNA	GSE69362	Agilent-069978 Arraystar Human CircRNA microarray V1	12	6 N/ 6 PDAC	[28,29]
	GSE79634	Agilent-069978 Arraystar Human CircRNA microarray V1	40	20 N/ 20 PDAC	[30]

N: normal tissue; PDAC: Pancreatic ductal adenocarcinoma.

**Table 2 cimb-47-00496-t002:** Significant hub genes identified from the ceRNA core network.

Hub Gene	Name	Function	Uniprot ID	Inhibitor	PubChem ID	Ref.
*ADAM12*	Disintegrin and metalloproteinase domain-containing protein 12	Functions in the development of preimplantation embryos	O43184	Abrine	160511	[50,51,52,53]
*MET*	MET Proto-Oncogene, Receptor Tyrosine Kinase	Affects angiogenesis, wound healing, morphogenesis, cell proliferation, cancer invasion, and survival	P08581	Acetaminophen	1983	[54,55,56]
*QKI*	Quaking Homolog, KH Domain Containing RNA Binding	Controls mRNA stability, translation, export of mRNAs from the nucleus, and pre-mRNA splicing	Q96PU8	Aristolochic acid I	2236	[57,58]
*SEC23A*	SEC23 Homolog A, COPII Coat Complex Component	Serving as a GTPase-activating protein	Q15436	1,2-Dimethylhydrazine	1322	[59,60]
*ZEB2*	Zinc Finger E-Box-Binding Homeobox 2	Activating Wnt/β-catenin signaling	O60315	Antimycin A	14957	[61,62,63]

**Table 3 cimb-47-00496-t003:** Repurposed drug candidates for potential PDAC treatment.

Drug	Drug Class	Mechanism of Action	Approval Status	Trial Number/Trial Status	Ref.
Vorinostat	Histone deacetylase (HDAC) inhibitor	Inhibits the enzymatic activity of HDAC	Approved	PAAD/NCT00948688/Phase I/II	[64]
				PA/NCT00983268/Phase I/II (Terminated)	
				PAAD/NCT02349867/Phase I	
Trichostatin A	Antifungal agent	Induces terminal differentiation, cell cycle arrest, and apoptosis in various cancer cell lines	Investigational	HM/NCT03838926/Phase I	[65,66]
Meclocycline sulfosalicylate	Antimicrobial agent	Inhibits protein synthesis in bacteria by binding to the 30S ribosomal subunit	Approved but discontinued	OM/NCT00385515/Phase II	[67]
Guanabenz acetate	Antihypertensive agent	Stimulates central alpha-2 adrenergic receptors, which reduces the release of norepinephrine	Investigational	MS/NCT02423083 /Phase I	[68]

PAAD: Pancreatic Adenocarcinoma, PA: Pancreatic Cancer, HM: Hematologic Malignancies, OM: Oral Mucositis, MS: Multiple Sclerosis.

## Data Availability

The dataset that was used in this study is publicly available at Gene Expression Omnibus (GEO Database) with the following links: GSE15471—https://www.ncbi.nlm.nih.gov/geo/query/acc.cgi?acc=GSE15471; GSE41368—https://www.ncbi.nlm.nih.gov/geo/query/acc.cgi?acc=GSE41368; GSE71989—https://www.ncbi.nlm.nih.gov/geo/query/acc.cgi?acc=GSE71989; GSE62165—https://www.ncbi.nlm.nih.gov/geo/query/acc.cgi?acc=GSE62165; GSE60978—https://www.ncbi.nlm.nih.gov/geo/query/acc.cgi?acc=GSE60978; GSE32678—https://www.ncbi.nlm.nih.gov/geo/query/acc.cgi?acc=GSE32678; GSE41369—https://www.ncbi.nlm.nih.gov/geo/query/acc.cgi?acc=GSE41369; GSE69362—https://www.ncbi.nlm.nih.gov/geo/query/acc.cgi?acc=GSE69362; GSE79634—https://www.ncbi.nlm.nih.gov/geo/query/acc.cgi?acc=GSE79634 (all links accessed on 10 February 2025).

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
