# Peer review of "The Construction of ceRNA Regulatory Network Unraveled Prognostic Biomarkers and Repositioned Drug Candidates for the Management of Pancreatic Ductal Adenocarcinoma"

_cimb, 2025, doi:10.3390/cimb47070496_

Round 1

Reviewer 1 Report

Comments and Suggestions for Authors

The authors took public databases to explore the probable oncogenic genes for biomarkers, but it did not uncover the new molecular signal. The genes of the hub genes have been well-characterized by other research, and the novelty of which emerges a concern for publication.  

  1. The resolution and labeling of figures requires to be improved, most figures samples are uncleared.
  2. Figure 6 showed the individual drug affinity to the target genes. However, this analysis did show the potential for target inactivation, but did not provide probable application. For example, Meclocycline sulfosalicylate, and anti-microbial agent, exhibited moderate binding affinities to most hub genes. What did the observation indicate? Because these genes were cancer-associated genes rather than bacterial infection-relevant genes. These findings confused the audiences for the authors’ observations.
  3. In Figure 7, a part of the candidate proteins’ expression was not detected by the IHC method, although the observation of the altered genes’ expression. The authors should interpret the inconsistent of these possible hub genes analyses, considering their expression were not enhanced in disease sections.
  4. Figures 1 and 2 showed DE-circRNAs and the proposed ceRNA network. However, the authors did not analyze the significance of circRNAs in the subsequent assays. Why the authors stop the circRNAs’ analyses?
  5. The authors proposed several miRNA-mRNA signaling and/or circRNA-miRNA -mRNA pathways. Did they confirm their observation with the published articles? It may accordingly evidence the significance of the current study.
  6. Despite the authors applied systematic exploration to uncover the probable oncogenic biomarker genes, and it remained the experimental validations for the findings. For examples, basic miRNA-mRNA interactions should be examined by Reporter assays and Western blot assays. The existing findings may not provide adequate supportive data for their theory.
Comments on the Quality of English Language

N/A

Author Response

Response to Reviewer- 1 comments:

We thank the Reviewer for valuable criticism and suggestions. We believe that these comments helped us to improve the quality of the manuscript. 

Comment: The resolution and labeling of figures requires to be improved, most figures samples are uncleared.

Response: Thank you for pointing this out. We have made necessary adjustments to the figures to fix resolution and readability.

Comment: Figure 6 showed the individual drug affinity to the target genes. However, this analysis did show the potential for target inactivation, but did not provide probable application. For example, Meclocycline sulfosalicylate, and anti-microbial agent, exhibited moderate binding affinities to most hub genes. What did the observation indicate? Because these genes were cancer-associated genes rather than bacterial infection-relevant genes. These findings confused the audiences for the authors’ observations.

Response: We appreciate the reviewer’s thoughtful comment regarding the antimicrobial nature of meclocycline sulfosalicylate and its predicted affinity for cancer-associated genes, which may have caused confusion. As demonstrated in Figure 6, meclocycline sulfocalicylate showed strong binding affinities when compared with inhibitors of the target genes.

We would like to clarify that the drug repositioning approach employed in our study was based on transcriptomic reversal signatures using the L1000CDS2 platform. This method does not rely on the traditional mechanism-of-action alignment but instead uses the ability of small molecules to reverse disease-specific gene expression patterns. Accordingly, the identification of meclocycline sulfosalicylate as a candidate compound was not based on its antimicrobial function per se, but rather on its predicted capacity to modulate the expression of oncogenic hub genes implicated in our network analysis.

Notably, several antimicrobial agents have recently been explored for their anticancer potential due to their secondary pharmacological properties, including anti-proliferative, anti-inflammatory, or epigenetic modulatory effects. Meclocycline sulfosalicylate’s moderate binding affinity to multiple cancer-associated targets in our analysis suggests a potential for repurposing, possibly with reduced systemic toxicity compared to conventional chemotherapeutics. As shown in functional enrichment (Figure 3), PDAC also shares similar DEGs with some infection-related diseases, including leishmaniasis, toxoplasmosis and Staphylococcus aureus infection. That is why it is not surprising to get antimicrobial agents for the potential treatment of cancers, since that can share common genes and pathways.

We have revised the manuscript to clarify this rationale in the Discussion section (see Lines 379-386), emphasizing the systems-level nature of our drug prioritization and the potential translational value of non-oncology compounds in novel therapeutic contexts.

Comment: In Figure 7, a part of the candidate proteins’ expression was not detected by the IHC method, although the observation of the altered genes’ expression. The authors should interpret the inconsistent of these possible hub genes analyses, considering their expression were not enhanced in disease sections.

Response: We appreciate the reviewer’s observation regarding the discrepancy between gene expression and protein-level detection in Figure 7. As correctly noted, two out of five candidate hub genes exhibited only moderate protein expression levels in IHC analyses, despite showing consistent upregulation at the transcriptomic level.

We would like to emphasize that our study employed a systems biology-driven approach that integrates transcriptomic data and network-based inference to prioritize candidate targets. Such computational predictions inherently reflect probabilistic associations rather than deterministic outcomes. Discrepancies between mRNA and protein expression are not uncommon, as they can result from various layers of post-transcriptional, translational, or post-translational regulation, as well as tissue-specific expression dynamics and technical limitations of detection methods such as IHC.

It is important to note that network biology aims to identify potential key regulatory nodes rather than guaranteeing uniform biological behavior across all molecular layers. If all predictions from omics-based modeling directly translated into clinically validated findings, the path to curing complex diseases such as cancer would be considerably less challenging.

Accordingly, we have added a clarification in the Discussion section (Lines 288–294) to acknowledge the observed inconsistency, and interpret it within the biological and methodological context, and reinforce the exploratory and hypothesis-generating nature of our approach.

Comment: Figures 1 and 2 showed DE-circRNAs and the proposed ceRNA network. However, the authors did not analyze the significance of circRNAs in the subsequent assays. Why the authors stop at the circRNAs’ analyses?

Response: As outlined in Figures 1 and 2, our initial aim was to construct a comprehensive ceRNA regulatory network in pancreatic ductal adenocarcinoma (PDAC) by integrating differentially expressed circRNAs, miRNAs, and mRNAs. This integrative approach was essential to uncover the multilayered regulatory mechanisms potentially driving disease pathogenesis.

While circRNAs were crucial in the construction of the ceRNA network and the elucidation of upstream regulatory pathways, our subsequent analyses focused on mRNAs for drug repositioning purposes. This decision was methodologically driven, as current drug repositioning platforms—such as L1000CDS2 and CMAP—operate primarily at the mRNA level and are not yet optimized for circRNA-based screening. Therefore, we prioritized mRNAs that are central nodes within the ceRNA network and involved in relevant signaling pathways as therapeutic targets.

Nevertheless, circRNAs played a foundational role in identifying these mRNAs within the regulatory network, and their expression profiles were considered in the mechanistic inference. We have added clarifying statements to the revised manuscript (see Lines 240–249) to better explain the rationale behind this analytical focus and to acknowledge the important, albeit indirect, contribution of circRNAs to the study’s therapeutic insights.

Comment: The authors proposed several miRNA-mRNA signaling and/or circRNA-miRNA -mRNA pathways. Did they confirm their observation with the published articles? It may accordingly evidence the significance of the current study.

Response: We appreciate the reviewer’s insightful suggestion regarding the validation of our proposed miRNA–mRNA and circRNA–miRNA–mRNA pathways with existing literature.

In constructing the PDAC-specific core regulatory network, we undertook an extensive literature review to evaluate the biological plausibility of the identified interactions. Several hub genes, including ADAM12, MET, THBS2, ZEB1, and ZEB2, have been previously implicated in PDAC progression, tumor invasiveness, and epithelial–mesenchymal transition (EMT). For example, ZEB1 and ZEB2 are established transcriptional repressors of E-cadherin and are well-documented drivers of EMT and metastasis in PDAC (Kurahara et al., 2012; doi:10.1002/jso.23020,  Krebs et al., 2017; doi:10.1038/ncb3513).

In parallel, several of the identified miRNAs, such as miR-200c-5p, miR-135b-5p, and miR-130b-5p, have been shown to regulate these hub genes in PDAC or related cancer types. miR-200c-5p, for instance, is a well-characterized suppressor of ZEB1/ZEB2, and its downregulation is associated with EMT activation in pancreatic cancer (Würdinger et al., 2008; https://doi.org/10.1016/j.ccr.2008.10.005). Similarly, miR-135b-5p has been linked to the regulation of MET and ADAM12, affecting invasion and proliferation in multiple cancers including PDAC (Li et al., 2019; doi:10.3892/ijo.2019.4732; Shao et al., 2025 ; doi.org/10.1016/j.ncrna.2025.02.005).

Although functional studies on circRNAs such as hsa_circRNA_102465 and hsa_circRNA_100904 remain limited, computational predictions and prior expression data suggest their potential involvement in gastrointestinal cancers and their capability to act as sponges for miRNAs relevant to PDAC regulatory circuits (Guo et al., 2022; doi.org/10.1097/JP9.0000000000000087).

We have now integrated this supporting information into the revised manuscript (Discussion, Lines 323-327 and 344–353), thereby strengthening the mechanistic validity and translational relevance of our findings.

Comment: Despite the authors applied systematic exploration to uncover the probable oncogenic biomarker genes, and it remained the experimental validations for the findings. For examples, basic miRNA-mRNA interactions should be examined by Reporter assays and Western blot assays. The existing findings may not provide adequate supportive data for their theory.

Response: As rightly noted, our current study employed a systems biology-guided in silico approach to identify putative oncogenic biomarkers and regulatory RNA interactions in pancreatic ductal adenocarcinoma (PDAC). This framework incorporated multi-omics data integration, network modeling, and transcriptome-based analyses to systematically prioritize candidate mRNAs, miRNAs, and circRNAs.

While we acknowledge that experimental assays such as luciferase reporter analysis or Western blotting would provide direct biochemical confirmation—particularly for key miRNA–mRNA interactions, the primary aim of this study was to construct a comprehensive predictive model that could serve as a foundation for future functional investigations. Notably, partial in silico validation was conducted through literature cross-referencing, target prediction consensus, and expression-based filtering from independent PDAC datasets.

We fully agree that further experimental validation is critical to translating these findings into clinically meaningful insights. Accordingly, we are in the process of preparing a follow-up research proposal aimed at functionally validating the most promising RNA–target interactions identified in this study through in vitro assays, including dual-luciferase reporter systems and Western blotting, in PDAC cell models.

We have included a paragraph in the revised Discussion section (Lines 440-450) to acknowledge this limitation and clarify our future directions for experimental validation.

Reviewer 2 Report

Comments and Suggestions for Authors

Remarks to authors

In this study, the authors explored the role of circular RNAs in pancreatic ductal adenocarcinoma (PDAC) by analyzing expression data from nine microarray datasets. A ceRNA network was constructed, identifying key hub genes (ADAM12, MET, QKI, SEC23A, ZEB2) linked to PDAC prognosis. Notably, these genes may serve as non-invasive biomarkers. Additionally, drug repositioning analysis suggested vorinostat, meclocycline sulfosalicylate, and trichostatin A as potential treatments. Overall, this study provides new insights into the regulation of PDAC carcinogenesis and finds some novel candidate biomarkers for prognosis. However, some key issues need to be addressed. My detailed comments are as follows.

  1. Please add more details and parameters in the figure legends (e.g., thresholds for fold changes, p-values). Moreover, clarify if any batch effect correction or normalization procedures were employed.
  2. In Figure 5, please clearly specify patient cohorts, sample sizes, and statistical tests used in survival analyses in the figure legends.
  3. Figure 7A lacks the scale bar.
  4. Figure 7B lacks sample sizes and the statistical test.
  5. As the drug studies are preliminary, hypothesis-generating findings requiring experimental validation, please discuss potential off-target effects, toxicity, and pharmacokinetics considerations for the proposed drugs, especially for repurposed compounds not yet approved for cancer treatment.
  6. Highlight what is novel in this study relative to existing literature. Are the particular circRNA-miRNA-mRNA interactions new? Is the drug repositioning approach innovative in PDAC?

Author Response

Response to Reviewer-2 comments:

We thank the Reviewer for the valuable criticism and suggestions. We believe that these comments helped us to improve the quality of our manuscript. 

Comment: Please add more details and parameters in the figure legends (e.g., thresholds for fold changes, p-values). Moreover, clarify if any batch effect correction or normalization procedures were employed.

Response: We appreciate the reviewer’s insightful comment. In response, all figure legends have been revised to address issues pointed out by the reviewer. Appropriate modifications were made to ensure that the interpretability of each figure meets publication standards. (see Lines 164-167)

Comment: In Figure 5, please clearly specify patient cohorts, sample sizes, and statistical tests used in survival analyses in the figure legends.

Response: We thank the reviewer for this important observation. In response, we have revised the legend of Figure 5 to provide detailed information regarding the patient cohorts, sample sizes, and statistical methodology used in the survival analyses. Specifically, we clarified that the survival curves were generated using the KM Plotter tool, based on a dataset comprising 1,640 samples, including 1,435 tumor tissues and 205 control tissues, across 20,433 genes, as described by Posta and Győrffy [38]. The patient data were obtained from publicly available GEO and ICGC datasets, with 94% of the tumor samples classified as pancreatic ductal adenocarcinoma (PDAC). Overall survival (OS) was assessed using the log-rank test, and hazard ratios (HRs) with 95% confidence intervals were calculated. Statistical significance was defined as log-rank p-values < 0.05. These revisions were made to ensure transparency and reproducibility in the interpretation of the survival analysis. (see Line 226-231)

Comment : Figure 7A lacks the scale bar.

Response: We thank the reviewer for the error on our part. The scale bars for all IHC figures have been added. (see figure 7A)

Comment: Figure 7B lacks sample sizes and the statistical test.

Response: We have revised the legend of Figure 7B to clearly indicate the source of the data (Human Protein Atlas), the statistical test used (two-tailed t-test), and the sample sizes involved in the analysis. Furthermore, the significance indicators (*, ***, and ****) have been explicitly defined, corresponding to p-values of <0.05, <0.001, and <0.0001, respectively. These additions were made to enhance the clarity of the figure (see Lines 296-299).

Comment: As the drug studies are preliminary, hypothesis-generating findings requiring experimental validation, please discuss potential off-target effects, toxicity, and pharmacokinetics considerations for the proposed drugs, especially for repurposed compounds not yet approved for cancer treatment.

Response: We appreciate the reviewer’s thoughtful and constructive comment. In response, we have expanded the discussion section to address key pharmacological considerations related to the proposed drug candidates.

Specifically, we acknowledged that although the identified compounds show therapeutic promise based on in silico predictions, their actual clinical efficacy depends on multiple pharmacokinetic and pharmacodynamic factors, including absorption, distribution, metabolism, excretion, and toxicity (ADMET). These parameters must be rigorously assessed in both in vitro and in vivo experimental models before clinical translation can be considered. Moreover, we have emphasized the need to evaluate potential off-target effects, particularly for repurposed compounds that are not currently approved for cancer treatment. Lastly, we highlighted the importance of experimentally validating the expression levels, regulatory interactions (such as those identified within the PDAC-specific ceRNA network), and functional roles of the hub genes through transcriptomic (e.g., qPCR) and proteomic (e.g., Western blotting, immunohistochemistry) approaches using clinically relevant PDAC samples. These additions collectively strengthen the translational relevance and reliability of our findings.

We have included a paragraph in the revised Discussion section (Lines 440-450) to acknowledge this limitation and clarify our future directions for experimental validation.

Comment: Highlight what is novel in this study relative to existing literature. Are the particular circRNA-miRNA-mRNA interactions new? Is the drug repositioning approach innovative in PDAC?

Response: We appreciate the reviewer’s comment regarding the novelty of our work. In response, we have revised the discussion to clearly highlight the innovative aspects of our study concerning the existing literature. Specifically, we emphasize that the circRNA–miRNA–mRNA regulatory axes identified in our PDAC-specific ceRNA network—including hsa_circRNA_102465–miR-135b-5p–ADAM12 and hsa_circRNA_100904–miR-200c-5p–ZEB1/ZEB2—represent previously uncharacterized interactions in the context of pancreatic ductal adenocarcinoma. These findings provide new insight into the post-transcriptional regulation of EMT-associated oncogenic drivers in PDAC.

Furthermore, we underscore the novelty of our transcriptomic reversal-based drug repositioning strategy, implemented via the L1000CDS2 platform. Unlike traditional repurposing approaches that rely on shared mechanisms of action, our method prioritizes drugs based on their ability to reverse disease-specific gene expression signatures. This allowed us to identify unconventional therapeutic candidates, including meclocycline sulfosalicylate and guanabenz acetate, in addition to the well-known HDAC inhibitors vorinostat and trichostatin A, none of which have been extensively explored in PDAC. These points have been integrated into the final paragraph of the discussion section to more explicitly communicate our study’s contributions to the field (see 451-468).

Round 2

Reviewer 2 Report

Comments and Suggestions for Authors

The authors have addressed all my concerns.

Author Response

Comment: The authors have addressed all my concerns.

Response: Thank you for the comment. We are pleased to hear that all the comments of the reviewer were addressed by us.